# Exploring the path to optimal diabetes care by unravelling the contextual factors affecting access, utilisation, and quality of primary health care in West Africa: A scoping review protocol

**Abdul-Basit Abdul-Samed**[1]*, **Ellen Barnie Peprah**[1], **Yasmin Jahan**[2],
**Veronika Reichenberger**[2], **Dina Balabanova**[2], **Tolib Mirzoev**[2], **Henry Lawson**[1], **Eric Odei**[3],
**Edward Antwi**[4], **Irene Agyepong**[1]

**1** Ghana College of Physicians and Surgeons, Accra, Ghana, **2** London School of Hygiene & Tropical Medicine, London, United Kingdom, **3** Korle Bu Teaching Hospital, Accra, Ghana, **4** Ghana Health Service, Accra, Ghana

* abdulbasitgunu@gmail.com

**Data Availability Statement:** No datasets were generated or analysed during the current study. All

## Abstract

### Background

The prevalence of diabetes in West Africa is increasing, posing a major public health threat. An estimated 24 million Africans have diabetes, with rates in West Africa around 2–6% and projected to rise 129% by 2045 according to the WHO. Over 90% of cases are Type 2 diabetes (IDF, World Bank). As diabetes is ambulatory care sensitive, good primary care is crucial to reduce complications and mortality. However, research on factors influencing diabetes primary care access, utilisation and quality in West Africa remains limited despite growing disease burden. While research has emphasised diabetes prevalence and risk factors in West Africa, there remains limited evidence on contextual influences on primary care. This scoping review aims to address these evidence gaps.

### Methods and analysis

Using the established methodology by Arksey and O'Malley, this scoping review will undergo six stages. The review will adopt the Preferred Reporting Items for Systematic Reviews and Meta-Analysis Extension for Scoping Review (PRISMA-ScR) guidelines to ensure methodological rigour. We will search four electronic databases and search through grey literature sources to thoroughly explore the topic. The identified articles will undergo thorough screening. We will collect data using a standardised data extraction form that covers study characteristics, population demographics, and study methods. The study will identify key themes and sub-themes related to primary healthcare access, utilisation, and quality. We will then analyse and summarise the data using a narrative synthesis approach.

relevant data from this study will be made available upon study completion.

**Funding:** This research was funded by the NIHR Global Health Research Centre for Non-Communicable Disease Control in West Africa using UK aid from the UK Government to support global health research. Funding was received from the National Institute for Health Research (NIHR) Global Health Research Centre (grant number NHIR203246) on Strengthening of Capacity for NCD Control in West Africa (Stop-NCD) (https://nihr.ac.uk/). Funding was awarded to IA and TM. The focus of this research is to strengthen the capacity for NCD control in West Africa. The views expressed in this publication are those of the author(s) and not necessarily those of the NIHR or the UK government. The funders had no role in study design, data collection and analysis, decision to publish, or preparation of the manuscript.

**Competing interests:** We declare no competing interests.

## Results

The findings and conclusive report will be finished and sent to a peer-reviewed publication within six months.

## Conclusion

This review protocol aims to systematically examine and assess the factors that impact the access, utilisation, and standard of primary healthcare services for diabetes in West Africa.

## Introduction

Sub-Saharan Africa (SSA) is currently grappling with the double burden of disease–communicable and non-communicable—which is impacting the lives of millions across the continent [1]. Diabetes, a non-communicable disease where the body either does not produce enough insulin or cannot use it effectively, is one of these [2].

Diabetes is a growing problem in SSA, and it is notably situated at the intersection of infectious and chronic disease processes and presents a significant threat to public health [3–5]. Approximately 24 million individuals are living with diabetes in Africa, and by 2045, this is projected to grow by 129% [6]. However, given the dearth of quality data and research on diabetes in SSA, prevalence estimates are uncertain as there is little evidence on the disease epidemiology [4, 5]. It is estimated that West Africa has a comparatively higher prevalence of diabetes compared to other African regions [7]. The prevalence of diabetes in countries like Ghana, Burkina Faso and Niger has been estimated at 2%– 2.6%, 1.7%– 2.1%, and 5.6%, respectively, based on data from the International Diabetes Federation (IDF) and the World Bank [8, 9].

The increasing burden of diabetes in SSA is reinforced by its nature, which puts young and older populations at risk. Among the identified types of diabetes–Type 1, Type 2, Gestational diabetes mellitus (GDM) and specific types of diabetes due to other causes [10], Type 2 diabetes accounts for the most cases in the region. By estimates, approximately 90% of people living with diabetes in SSA have Type 2 diabetes. Additionally, in the Africa Region of IDF, around 52 million people (aged 20–79) are estimated to have Impaired Glucose Tolerance (IGT), significantly increasing their risk of developing Type 2 diabetes [8]. As of 2021, 59,500 cases of Type 1 diabetes were diagnosed in SSA and the prevalence of GDM was estimated at 13% by the IDF.

Given the diverse cultural fabric of SSA, the quality of diabetes care is impacted by unique challenges. Cultural beliefs, misconceptions, and stigma surrounding diabetes, among others, can hinder effective care, patient-provider relationships, and treatment adherence [11, 12]. Primary health care (PHC), which serves as the basic level of care for managing diseases, including diabetes, is pivotal in addressing these challenges and delivering comprehensive care. According to the WHO, PHC is widely recognised as the most inclusive, equitable, and cost-effective approach to achieving universal health coverage [13, 14]; however, many parts of the developing world, including SSA, face obstacles to access, utilisation, and quality of primary health care services. Various systemic and contextual factors influence healthcare delivery in these regions, resulting in disparities and gaps in diabetes care [15]. Addressing these challenges is crucial to providing patient-centred care that is culturally sensitive and respects the individual's beliefs, ideas, concerns, expectations and preferences. This will consequently lead to improved health outcomes for people living with diabetes.

## Access and utilisation

Access to PHC services is a prerequisite for effectively managing diabetes, preventing complications, and improving patient outcomes. Despite this, access to healthcare in Sub-Saharan Africa is estimated to be 42.56%, highlighting the critical need to eliminate barriers that limit access to essential healthcare services [16]. According to a review by Bresick et al., the state and performance of PHC in SSA need to be adequately measured, indicating a gap in the current research landscape [17].

Whereas "access to healthcare" can be described as the opportunity to fulfil healthcare needs, it encompasses various dimensions, including approachability, acceptability, affordability, availability and appropriateness [18]. Across the various dimensions, many factors influence access to and utilisation of care in the developing world. These include inadequate healthcare infrastructure, limited resources, inadequate healthcare workforce, sociocultural beliefs and practices, and economic disparities [18, 19].

In Lower and Middle-Income Countries (LMICs), including those in West Africa, geographic barriers play a pivotal role in the underutilisation of primary healthcare for diabetes. The literature suggests a significant unmet need for diabetes care in parts of SSA, indicating that health services utilisation may be suboptimal [20]. Remote and underserved communities often lack easy access to healthcare facilities, resulting in delayed diagnoses, limited follow-up care, and compromised disease management [21]. Additionally, cultural beliefs, traditional healing practices, and misconceptions surrounding diabetes can impact healthcare-seeking behaviour, leading to the underutilisation of orthodox medicine. The literature shows that cultural beliefs and practices can sometimes impact the decision to seek medical care for diabetes. Examples include beliefs that witchcraft, curses, or spiritual forces cause diabetes, the stigma that diabetes is a disease only affecting the wealthy, and the preference for herbal and traditional medicines over modern antidiabetic medications due to perceived superiority or safety [1, 4, 12]. The utilisation patterns of people living with Non-communicable Diseases (NCDs) are further influenced by various factors, including awareness or knowledge of diabetes, attitudes towards seeking care and availability of services. The burden of diabetes further compounds the strain on already overburdened healthcare systems, necessitating innovative solutions to promote the utilisation of available care.

Access to and utilisation of healthcare services are closely interconnected, and certain factors can impact both aspects simultaneously. While it may be challenging to separate these factors, it is essential to understand their distinct influences:

**Financial Limitations:** Accessing healthcare services can become challenging when financial resources are limited. High out-of-pocket expenses, absence of health insurance coverage, and the inability to afford transportation expenses can prevent people from accessing healthcare facilities. Furthermore, financial limitations can lead to underutilisation of services as some individuals may avoid seeking care or delay treatment due to cost concerns [5].

**Healthcare Infrastructure**: Access and utilisation of quality healthcare can be hindered by insufficient infrastructure, such as poorly equipped facilities and inadequate medical supplies. When healthcare centres are understaffed or lack necessary equipment, individuals may face difficulties in obtaining proper care, which could result in reduced utilisation. These challenges may lead people to seek alternative sources of care or even forego treatment altogether [5, 21].

**Healthcare Workforce Shortages**: Insufficient staffing levels in healthcare, including a shortage of doctors and nurses, can negatively affect patient access and utilisation. When

healthcare facilities are understaffed, patients may experience longer wait times and reduced access to necessary medical attention. This shortage can make it difficult for people to schedule timely appointments or receive comprehensive care, ultimately affecting their ability to utilise healthcare services [22].

**Sociocultural Beliefs and Practices**: Social and cultural factors significantly impact people's access to and use of healthcare services. Cultural beliefs and norms may discourage individuals from seeking proper medical care, which results in limited access. Additionally, cultural practices and preferences may influence the utilisation of healthcare services, with some people opting for traditional healers or home remedies instead of professional medical care [11].

## Quality

The World Health Organization defines the delivery of quality care as the extent to which health services improve the chances of achieving desired health outcomes for both individuals and populations. Achieving this requires a comprehensive approach that considers various aspects of care. This ensures that PHC services are safe, efficient, effective, patient-centred, integrated, equitable, and timely [23]. In SSA, several barriers present significant challenges to achieving and upholding the necessary standards of care [15, 24]. Various factors identified in the literature which influence the quality of care include resource limitations, health worker training and capacity building, adherence to clinical guidelines and patient-provider communication [5, 22]. Managing diabetes is a continuous process that requires consistent care and support to prevent complications and maintain good health. To ensure effective management of the condition, it is essential to identify and address the contextual factors that influence PHC for diabetes to help provide ongoing care and support, prevent complications, and promote overall well-being [25].

## Rationale

People with chronic conditions such as diabetes need long-term involvement with the healthcare delivery system, coordinated contributions from a variety of health professionals, access to necessary medications, and monitoring systems that are well-integrated into a system that encourages patient empowerment [26].

The standard of care provided to diabetes patients have an impact on long-term results with good diabetic management that lowers the incidence of cardiovascular and micro-vascular complications [27, 28].

However, despite the enormous progress that has been made in the management and treatment of diabetes in recent years, many patients still do not get the most favourable outcomes and continue to suffer from life-threatening complications that reduce both their lifespan and quality of life. It has been argued that the primary level of health care is best suited to the care of person with the chronic condition, and it is challenging to provide chronic care at primary level specially in Low- and -Middle income areas like West Africa. Further, the large number of persons with diabetes who fail to meet clinical target levels demonstrates that there is still a significant gap between knowledge, comprehension, identifying contextual factors and effective healthcare [29]. Lau et al. also showed that many multilevel contextual influences related to the policy environment, organizational context, health professionals in primary care, and intervention characteristics affect the ability of primary care delivery systems and their actors to change [30]. Therefore, it is essential to synthesise the available data on contextual factors influencing primary care access, quality, and

utilisation to have a thorough understanding of the contextual factors that support or hinder effective PHC for diabetes in West Africa. Additionally, it is equally important to comprehend the contextual implications on patients' access to and use of PHC services, treatment outcomes, and quality of care, although most research has focused on the prevalence and risk factors for diabetes [6, 27, 28]. Till now, evidence on contextual factors focusing on access, utilisation and quality of primary care among diabetes patients is still limited in West Africa. Hence this scoping review protocol aims to synthesise existing literature and identify the gaps on the contextual factors influencing access, utilisation, and quality of primary healthcare for diabetes in West Africa, and how and why they work. By identifying barriers, gaps, and successful strategies, this review may help to improve diabetes primary healthcare, improve patient outcomes, promote health equity, and develop further interventions that will ultimately help to reduce the burden of diabetes in West Africa.

## Objectives

1. To identify and synthesise existing literature on the contextual factors influencing access, utilisation, and quality of primary healthcare for diabetes in West Africa, and how and why they work.

2. To identify gaps in the current literature for further research to inform intervention to improve access, utilisation, and quality of primary healthcare for diabetes in West Africa.

## Conceptual framework for the scoping review

To guide our review we will draw on a conceptual framework to rigorously examine multi-level contextual factors shaping primary care services for diabetes patients in the region. Context refers to the surroundings or mileau in which a particular event or phenomenon is being studied; context is increasingly recognised as a crucial factor shaping health and healthcare [31]. For the purpose of this scoping review, context refers to the diverse physical, social, cultural, economic, political, and health systems environments that influence opportunities for optimal diabetes prevention and care. Contextual factors interact across macro, meso, and micro levels [32, 33] as illustrated in Fig 1. Macro-level elements include cultural norms, health policies, and demographic changes. Meso-level factors consist of community resources, healthcare facilities, staffing, and medical supply chains. Micro-level factors encompass individual attitudes, knowledge, interpersonal relationships, and living conditions. These contextual factors interact to impact PHC access, utilisation, and quality in West Africa, acting as both barriers and enablers.

## Methodology

A preliminary literature scan indicates little published evidence on PHC for diabetes in West Africa; a scoping review is an appropriate method for mapping out any available evidence in this area. The methodology of the protocol for the scoping review will adhere to the framework created by Arksey and O'Malley [34, 35]. The methodology will therefore include the following stages:

1. Formulating the research question

2. Identifying relevant literature

3. Study Selection

**Fig 1. Contextual factors at macro, meso and micro levels.**

4. Charting the data

5. Collating, summarising and reporting the results

6. Peer consultation, feedback and validation

In order to conduct and report the scoping review in a well-organised manner, we will be utilising the recommended tools by the Joana Briggs Institute [36], namely the Population, Concept, and Context (PCC) framework and the PRISMA-ScR (Preferred Reporting Items for Systematic Reviews and Meta-Analyses Extension for Scoping Reviews) guidelines and checklist.

## Stage 1—Identifying the review question

Through an iterative process and extensive consultation with the research team, the research question for the review has been carefully identified as follows:

What contextual factors influence primary healthcare access, utilisation and quality for diabetes in West Africa, and how and why do they work and what are the gaps in the current literature?

## Stage 2—Identifying relevant studies

To identify the studies, keyword searches of academic databases will be conducted. A comprehensive search strategy will be developed to identify relevant literature from four selected electronic databases, i.e. PubMed, Google Scholar, African Journals Online (AJOL), and CAIRN INFO (for French papers). We selected these four databases because together, they provide comprehensive coverage of medical, public health, African-centered, and French language research related to the biomedical and health sciences in West Africa. In addition, grey literature sources such as the World Health Organization (WHO) website, the NCD Alliance and Ministries of Health where possible. This will be done with the help of a librarian. The search strategy will be developed using a combination of keywords and alternative vocabulary terms derived from the key concepts of the study (S1 Table). A complete search strategy, applied to PubMed (S2 Table), will be subsequently developed and then adapted to other databases. A sample search strategy on PubMed is provided in S3 Table. The search results will be managed using reference manager software (for example, EndNote, Zotero).

## Stage 3—Study selection

The Population, Concept and Context Study framework shown in Table 1 will align the inclusion of studies. The framework will be used to guide title and abstract screening. The inclusion and exclusion criteria shown in Table 2 will initially guide the review. The criteria will iteratively undergo further refinement as the reviewers gain a better understanding of the literature's scope.

A decision-making flowchart (S1 Fig) will aid the study selection process.

The abstracts will subsequently be reviewed for any mention of factors that influence access, utilisation or quality of PHC in adults with diabetes in West Africa. Due to its convenience, reviewers will use Rayyan software to independently screen the articles based on the inclusion and exclusion criteria. After the initial abstract review is completed, the full texts will be accessed for final screening and data extraction. To ensure the reliability and validity of the source selection process, two reviewers will independently perform source selection at both the title/abstract screening and full-text screening stages. Any disagreements among the reviewers will be resolved through a joint review and discussion of the disagreement by all reviewers guided by the decision flowchart (S1 Fig) to determine eligibility through consensus. A pilot test will be undertaken on the process to ensure effective source selection by randomly selecting a 5% sample of titles/abstracts. The entire team will screen them according to the eligibility criteria and discuss any discrepancies. The selection process will continue until a 75% agreement or higher is reached. A record of the progress of the source-selection process will be maintained for all full-text articles retrieved, including details of included and excluded sources. Any excluded sources will have reasons clearly stated for why they were excluded.

**Table 1. Population, concept and context framework.**

| PCC Element | Definition/ Keywords |
| --- | --- |
| Population | Adults [Age 18 and above] living with diabetes [Type 1or Type 2] |
| Concept | • Access<br>• Utilisation<br>• Quality<br>• Primary Health Care |
| Context | West Africa |

**Table 2. Inclusion and exclusion criteria.**

|  | Inclusion | Exclusion |
|---|---|---|
| **Year of publication** | 2000 | N/A |
| **Language** | English, French | Other Languages |
| **Population** | Adults with diabetes | People with gestational diabetes |
| **Concept** | Access | N/A |
|  | Utilisation |  |
|  | Quality |  |
|  | PHC |  |
|  | Factors |  |
| **Context** | West Africa | Regions outside West Africa |
| **Study design** | • Qualitative, Quantitative [if any], Meta-analysis, Social Science Case studies/reports<br>• Multi-country studies which include West African countries | • Studies that do not contain information on factors affecting access, utilisation and quality of PHC<br>• Abstracts<br>• Commentaries<br>• Opinions<br>• Scoping or Systematic review protocols |

NA–Not Applicable

## Stage 4—Charting the data

We will use a data charting/extraction tool (Table 3) recommended by the Joana Brigg's Institute [37] to capture relevant information from each study.

## Stage 5—Collating, summarising and reporting the results

To ensure transparency and clarity in reporting findings, the review team will be utilising the PRISMA (Preferred Reporting Items for Systematic Reviews and Meta-Analyses) flow diagram (Fig 2) along with the PRISMA-ScR and PRISMA-P checklist (S4 and S5 Tables) [38, 39].

**Table 3. Data extraction tool.**

| |
|---|
| **Author and Date** |
| **Title** |
| **Aim** |
| **Study Setting** |
| **Study Population** |
| **Sampling Method** |
| **Study Design** |
| **Data Collection Method** |
| **Data Analysis** |
| **Conclusion** |
| **Outcome** |
| **Most relevant Findings** |
| **Comment** |

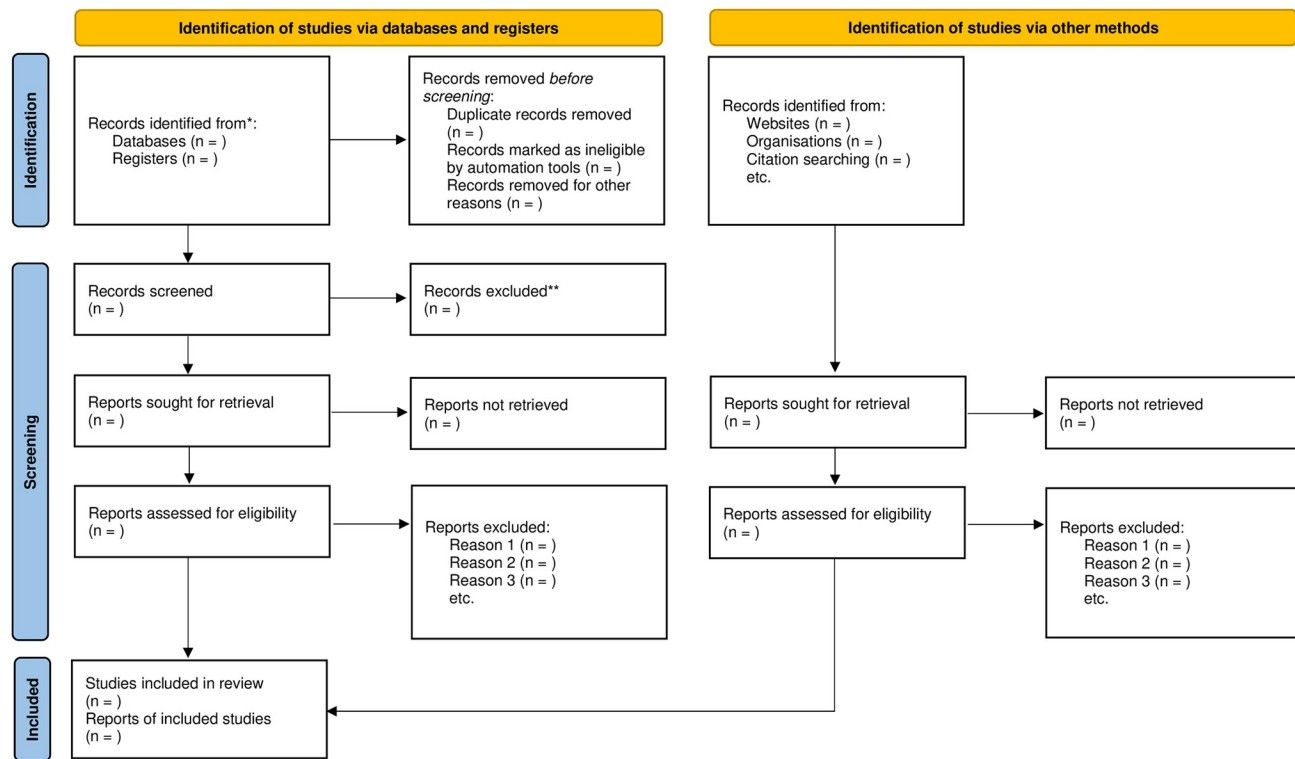

**Fig 2. PRISMA flowchart.**

The data extracted from the included studies will be organized and categorized based on key themes and concepts from the conceptual framework developed to guide this review on access, utilisation and quality of diabetes care. The framework will provide an initial coding structure, however an inductive approach will also be utilized to allow for additional codes and themes to emerge directly from the data.

Factors influencing access, utilization, and quality will be identified through careful reading and coded using a qualitative data analysis software. The codes will capture barriers and facilitators at the patient, provider, health system and contextual levels. Pattern analysis will be conducted to examine relationships and associations between codes Data segments that address the research questions will be pooled. Qualitative metasynthesis will be used to integrate key findings across studies and generate new interpretations about access and quality of diabetes care in West Africa. Once the results have been collated, they will be summarised clearly and concisely. Reporting the results will involve the creation of a comprehensive narrative that presents the synthesised findings in a logical and structured manner. The narrative will summarise the key findings, including the prevalence and distribution of the identified factors and any variations observed across different countries or regions within West Africa. The results will be discussed in light of the research questions and objectives, highlighting the implications for policy, practice, and future research.

### Stage 6 –Peer consultation for feedback and validation

As we work on the project, we will collaborate with subject matter specialists to gain their insights and receive feedback. In stage 2, a skilled librarian will assess the search strategy through a peer review process to ensure that it is comprehensive and relevant. In addition, we plan to utilise our network to engage more experts in the field so that all relevant discoveries are reported with precision and clarity. Throughout stages 3 to 5, we will maintain communication with these experts to gather their recommendations for further reading and to receive feedback on data extraction.

## Discussion

This scoping review protocol seeks to methodically investigate and evaluate the elements that affect the accessibility, usage, and quality of primary healthcare for diabetes in West Africa. The goal is to carefully search and analyse relevant studies to gain insights into the factors that affect diabetes care in the region. A thorough search strategy to select the most informative studies will be employed. The findings of this scoping review will help to improve understanding of the current landscape of diabetes care in West Africa and provide valuable information to improve the delivery and outcomes of PHC for individuals with diabetes.

### Strengths and limitations

This review will follow an established framework and use a comprehensive systematic search to identify and analyse existing literature. Our search strategy is peer-reviewed, and we will consult with experts to guide our review and include relevant grey literature. The heterogeneity and limitations of the individual studies included in the scoping review may also make it challenging to draw definitive conclusions or generalise findings. It is worth noting that the scope of this evaluation is limited to studies conducted in West Africa, which may restrict the applicability of the findings to other regions or populations. Additionally, the review is confined to specific languages [English and French], which may also affect the findings.

### Dissemination

The findings will be disseminated through various channels, including peer-reviewed publications, policy briefs, conferences, and active engagement with relevant stakeholders. The goal is to inform evidence-based healthcare practices, shape policy interventions, and foster meaningful improvements in diabetes care for diabetes in West Africa, informed by evidence of factors influencing PHC for diabetes.

## Supporting information

**S1 Fig. Decision flowchart.**
(TIF)

**S1 Table. Key words combination chart.**
(DOCX)

**S2 Table. Search terms for PubMed.**
(DOCX)

**S3 Table. Sample search for PubMed.**
(DOCX)

**S4 Table. PRISMA-ScR checklist.**
(DOCX)

**S5 Table. PRISMA-P checklist.**
(DOC)

## Author Contributions

**Conceptualization:** Abdul-Basit Abdul-Samed, Dina Balabanova, Tolib Mirzoev, Eric Odei, Irene Agyepong.

**Funding acquisition:** Tolib Mirzoev, Irene Agyepong.

**Methodology:** Abdul-Basit Abdul-Samed.

**Writing – original draft:** Abdul-Basit Abdul-Samed, Eric Odei.

**Writing – review & editing:** Ellen Barnie Peprah, Yasmin Jahan, Veronika Reichenberger, Dina Balabanova, Tolib Mirzoev, Henry Lawson, Edward Antwi, Irene Agyepong.

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
