## [Decision Letter · Decision Letter 0]

28 Sep 2023

PONE-D-23-22733Exploring the path to optimal diabetes care by unravelling the contextual factors affecting access, utilisation, and quality of primary health care in West Africa: a scoping review protocolPLOS ONE

Dear Dr. Abdul-Basit Abdul-Samed

Thank you for submitting your manuscript to PLOS ONE. After careful consideration, we feel that it has merit but does not fully meet PLOS ONE’s publication criteria as it currently stands. Therefore, we invite you to submit a revised version of the manuscript that addresses the points raised during the review process. The reviewers have raised important points pertaining the manuscript as it relates to a study protocol. 

Please submit your revised manuscript within two weeks. If you will need more time than this to complete your revisions, please reply to this message or contact the journal office at plosone@plos.org. Please include the following items when submitting your revised manuscript:A rebuttal letter that responds to each point raised by the academic editor and reviewer(s). You should upload this letter as a separate file labeled 'Response to Reviewers'.A marked-up copy of your manuscript that highlights changes made to the original version. You should upload this as a separate file labeled 'Revised Manuscript with Track Changes'.An unmarked version of your revised paper without tracked changes. You should upload this as a separate file labeled 'Manuscript'.

We look forward to receiving your revised manuscript.

Kind regards,

Mergan Naidoo, PhD

Academic Editor

PLOS ONE

Journal Requirements:

“This research was funded by the NIHR Global Health Research Centre for Non-Communicable Disease Control in West Africa using UK aid from the UK Government to support global health research. The views expressed in this publication are those of the author(s) and not necessarily those of the NIHR or the UK government.”

“I.A. and T.M. received funding from the National Institute for Health Research (NIHR) Global Health Research Centre (grant number NHIR203246). The funder for this grant is the NIHR Global Health Research Centre on Strengthening of Capacity for NCD Control in West Africa (Stop-NCD) (https://nihr.ac.uk/). The focus of this research is to strengthen the capacity for NCD control in West Africa.”

Reviewers' comments:

Reviewer's Responses to Questions

**Comments to the Author**

1. Does the manuscript provide a valid rationale for the proposed study, with clearly identified and justified research questions?

Reviewer #1: No

Reviewer #2: Yes

Reviewer #3: Yes

2. Is the protocol technically sound and planned in a manner that will lead to a meaningful outcome and allow testing the stated hypotheses?

Reviewer #1: No

Reviewer #2: Partly

Reviewer #3: Partly

3. Is the methodology feasible and described in sufficient detail to allow the work to be replicable?

Reviewer #1: No

Reviewer #2: Yes

Reviewer #3: Yes

4. Have the authors described where all data underlying the findings will be made available when the study is complete?

Reviewer #1: No

Reviewer #2: Yes

Reviewer #3: Yes

5. Is the manuscript presented in an intelligible fashion and written in standard English?

Reviewer #1: No

Reviewer #2: Yes

Reviewer #3: Yes

6. Review Comments to the Author

You may also provide optional suggestions and comments to authors that they might find helpful in planning their study.

Reviewer #1: Dear Author,

Though your topic of study is sound, what you submitted doesn't sound like the manuscript for publication but rather a protocol for other use.

This mistake disqualifies your submission from my side.

Reviewer #2: Dear authors,

Thank you for your efforts.

The manuscript tackles a prevalent and a preventable disease.

The rationale of the manuscript is acceptable.

However, it would be better to see the proposed results in the final manuscript.

I strongly suggest the authors finalise the results and present them in the manuscript with all these backgrounds and methodologies.

Regards

Reviewer #3: This is a review of a manuscript entitled “Exploring the path to optimal diabetes care by unravelling the contextual factors affecting access, utilization, and quality of primary health care in West Africa: a scoping review protocol”

It is generally quite well-written, and I have a few comments to further improve it.

Please kindly consider some comments below

Abstract

Lines 39

I don't see stats in background

This would support that diabetes is a pressing public health challenge in West Africa

Also, it would be good if the background stated what is known about the research question (although very briefly, but it is there) and what is unknown (it is missed).

Introduction

Lines 82-85

It’s enough to mention what diabetes is, I don’t think it’s worth going into details, it overloads the introduction

Line 96

I'm wondering if there are estimates for type 1 diabetes and gestational diabetes?

It would be good to include them here, the authors only mention estimates for type 2 diabetes

Line 132

Would be good to say more about cultural beliefs that can impact healthcare-seeking behavior, do authors know from the literature which cultural beliefs and practices among people living with diabetes may impact the approach to seek health care?

Stage 2 - Identifying Relevant Studies

Line 235

Can authors clarify why these four electronic databases were selected?

Line 265-266

It seems quite subjective if an added review will resolve any disagreements among the reviewers

Line 326

Limitations of included studies may also play a role and may make it challenging to draw conclusions

7. PLOS authors have the option to publish the peer review history of their article (what does this mean?). If published, this will include your full peer review and any attached files.

Reviewer #1: **Yes: **Adeloye Amoo Adeniji (MBBS;MMed;FCFP;FACRRM)

Reviewer #2: No

Reviewer #3: No

---

## [Author Response · Author response to Decision Letter 0]

26 Oct 2023

Abdul-Basit Abdul-Samed

Ghana College of Physicians and Surgeons

Accra, Ghana

abdulbasitgunu@gmail.com

+233558059723

26/10/2023

Mergan Naidoo, PhD

Academic Editor

PLOS ONE

Dear Editor,

Thank you for considering our paper titled "Exploring the path to optimal diabetes care by unravelling factors affecting access, utilisation, and quality of primary health care in West Africa: a scoping review protocol " for publication in your esteemed journal. We appreciate the time and effort the reviewers have dedicated to providing feedback on our work. Their comments and suggestions have been immensely valuable.

We have carefully studied the reviewers' comments and have made revisions to the manuscript accordingly. In this rebuttal letter, we have responded to each of the points raised by the reviewers and outlined the changes made (indicating the specific lines in the revised manuscript with track changes). We believe these revisions have helped improve the quality and clarity of our manuscript significantly.

Reviewer #1 comments 

“Though your topic of study is sound, what you submitted doesn't sound like the manuscript for publication but rather a protocol for other use. This mistake disqualifies your submission from my side.”

Response to reviewer #1

Thank you. The submitted document is a protocol paper, and it is deliberated and submitted for publication as a protocol paper rather than as a completed research /review manuscript. This is reflected in the title “Exploring the path to optimal diabetes care by unravelling the contextual factors affecting access, utilisation, and quality of primary health care in West Africa: a scoping review protocol”

We also indicated in the cover letter and application that we were submitting a protocol paper. Protocol papers are useful for laying out the methods for a planned review in a transparent manner before embarking on the review itself. Publishing protocols can help improve the quality of the eventual review and allow readers to provide feedback on the proposed methods. 

Reviewer #2 comments

“Dear authors, Thank you for your efforts. The manuscript tackles a prevalent and a preventable disease. The rationale of the manuscript is acceptable. However, it would be better to see the proposed results in the final manuscript. I strongly suggest the authors finalise the results and present them in the manuscript with all these backgrounds and methodologies. Regards”

Response to Reviewer 2

Thank you for taking the time to review our manuscript and provide your feedback. We appreciate you finding the rationale acceptable and agreeing that diabetes is a critical public health issue that needs to be addressed, especially in regions like West Africa. The results are not yet available since the scoping review is now being conducted. Protocol papers are useful for laying out the methods for a planned review in a transparent manner before embarking on the review itself. Publishing protocols can help improve the quality of the eventual review and allow readers to provide feedback on the proposed methods. 

We appreciate you highlighting the importance of sharing the final results when the planned review is completed. We agree that disseminating the findings through peer-reviewed publication will be impactful. As outlined in our protocol methods, we plan to finish conducting the scoping review and summarise the results in a manuscript for publication within the next six months. This future manuscript will contain the results and findings from synthesising the data extracted from studies meeting our inclusion criteria. 

Reviewer #3 comments

Reviewer Comment

“This is a review of a manuscript entitled “Exploring the path to optimal diabetes care by unravelling the contextual factors affecting access, utilization, and quality of primary health care in West Africa: a scoping review protocol” “It is generally quite well-written, and I have a few comments to further improve it. Please kindly consider some comments below

Reviewer Comment

Abstract. Lines 39

I don't see stats in background. This would support that diabetes is a pressing public health challenge in West Africa. Also, it would be good if the background stated what is known about the research question (although very briefly, but it is there) and what is unknown (it is missed).”

Response

Thank you for highlighting the importance of including statistics in the background section. We have added prevalence statistics and the relevant references to the background section as follows (Lines 38 – 51):

The prevalence of diabetes in West Africa is increasing, posing a major public health threat. An estimated 24 million Africans have diabetes, with rates in West Africa around 2-6% and projected to rise 129% by 2045 (WHO). Over 90% of cases are Type 2 diabetes (IDF, World Bank). As diabetes is ambulatory care sensitive, good primary care is crucial to reduce complications and mortality. However, research on factors influencing diabetes primary care access, utilization and quality in West Africa remains limited despite growing disease burden. While research has emphasised diabetes prevalence and risk factors in West Africa, there remains limited evidence on contextual influences on access, utilisation and quality of primary care. This scoping review aims to address these evidence gaps.

Reviewer comment

“Introduction. Lines 82-85

It’s enough to mention what diabetes is, I don’t think it’s worth going into details, it overloads the introduction”

Response

Thank you for the suggestion. We have tightened the text and removed unnecessary detail. 

Reviewer comment

“Line 96. I'm wondering if there are estimates for type 1 diabetes and gestational diabetes?

It would be good to include them here, the authors only mention estimates for type 2 diabetes”

Response

While data on the estimates of type 1 diabetes and GDM in West Africa are limited, the following information from the literature, with the reference, is included to address this helpful suggestion (Lines 105 – 107):

As of 2021 59,500 cases of Type 1 diabetes were diagnosed in SSA and the prevalence of GDM was estimated at 13% by the IDF.

Reviewer Comment

“Line 132

Would be good to say more about cultural beliefs that can impact healthcare-seeking behavior, do authors know from the literature which cultural beliefs and practices among people living with diabetes may impact the approach to seek health care?”

Response: 

We have added the following text (with references) to address this comment (Lines 144 – 149):

The literature shows that cultural beliefs and practices can sometimes impact the decision to seek medical care for diabetes. Examples include the belief that witchcraft, curses, or spiritual forces cause diabetes, the stigma that diabetes is a disease only affecting the wealthy, and the preference for herbal and traditional medicines over modern antidiabetic medications due to perceived superiority or safety.

Reviewer Comment

“Stage 2 - Identifying Relevant Studies. Line 235. Can authors clarify why these four electronic databases were selected?”

We selected these four databases because together, they provide comprehensive coverage of medical, public health, African-centered, and French language research related to the biomedical and health sciences in West Africa. Specifically:

• PubMed: As a database of biomedical and life sciences journal literature, it provides extensive coverage of health and medical research, including diabetes care in West Africa.

• Google Scholar: It indexes a wide range of peer-reviewed literature across disciplines and sources. 

• African Journals Online (AJOL): This database contains leading journals published in Africa and provides a way to comprehensively search African-specific literature.

• CAIRN Info: This French database allows us to search Francophone literature and identify studies published in French from West Africa.

Reviewer comment

“Line 265-266

It seems quite subjective if an added review will resolve any disagreements among the reviewers”

Response

Thank you for highlighting this point about the subjectivity of using an additional reviewer to resolve disagreements during the study screening process. We agree that just adding another reviewer does not guarantee resolving disagreements in a completely objective manner.

Upon reflection, we agree the current proposal makes the process potentially subjective. To address this, we propose removing the mention of an "added reviewer" and revising the sentence as follows (Lines 332 – 327):

"Any disagreements among the reviewers will be resolved through a joint review and discussion of the disagreement by all reviewers guided by the decision flowchart to determine eligibility through consensus.

Reviewer comment

“Line 326

Limitations of included studies may also play a role and may make it challenging to draw conclusions”

We agree that the limitations of the individual studies included in our scoping review can also affect our ability to draw firm conclusions. To address this, we have added the following to the Limitations section:

"The heterogeneity and limitations of the individual studies included in the scoping review may also make it challenging to draw definitive conclusions or generalise findings."

Adding this limitation will help preemptively set reasonable expectations for the scoping review results. 

Additional comments

Kindly note the improvement of the data analysis section as follows (Lines 368 – 378):

The data extracted from the included studies will be organised and categorized based on key themes and concepts from the conceptual framework developed to guide this review on access, utilisation and quality of diabetes care. The framework will provide an initial coding structure; however, an inductive approach will also be utilised to allow for additional codes and themes to emerge directly from the data. 

Factors influencing access, utilisation, and quality will be identified through careful reading and coded using qualitative data analysis software. The codes will capture barriers and facilitators at the patient, provider, health system and contextual levels. Pattern analysis will be conducted to examine relationships and associations between codes Data segments that address the research questions will be pooled. Qualitative meta-synthesis will be used to integrate key findings across studies and generate new interpretations about access and quality of diabetes care in West Africa. 

Thank you again for considering our work and providing us the opportunity to improve it through peer review. We highly appreciate your time and effort.

Sincerely,

Abdul-Basit Abdul-Samed

---

## [Editor Report · Decision Letter 1]

13 Nov 2023

Exploring the path to optimal diabetes care by unravelling the contextual factors affecting access, utilisation, and quality of primary health care in West Africa: a scoping review protocol

PONE-D-23-22733R1

Dear Dr. Abdul-Basit Abdul-Samed

We’re pleased to inform you that your manuscript has been judged scientifically suitable for publication and will be formally accepted for publication once it meets all outstanding technical requirements.

Kind regards,

Mergan Naidoo, PhD

Academic Editor

PLOS ONE
---

## [Editor Report · Acceptance letter]

8 May 2024

PONE-D-23-22733R1 

PLOS ONE

Dear Dr. Abdul-Samed, 

I'm pleased to inform you that your manuscript has been deemed suitable for publication in PLOS ONE. Congratulations! Your manuscript is now being handed over to our production team.

Kind regards, 

on behalf of

Professor Mergan Naidoo 

Academic Editor

PLOS ONE